# FREA: Feasibility-Guided Generation of Safety-Critical Scenarios with Reasonable Adversariality

**Keyu Chen**[1]**, Yuheng Lei**[2]**, Hao Cheng**[1]**, Haoran Wu**[1]**, Wenchao Sun**[1]**, Sifa Zheng**[1]

[1]School of Vehicle and Mobility, Tsinghua University
[2]The University of Hong Kong

**Abstract:** Generating safety-critical scenarios, which are essential yet difficult to collect at scale, offers an effective method to evaluate the robustness of autonomous vehicles (AVs). Existing methods focus on optimizing adversariality while preserving the naturalness of scenarios, aiming to achieve a balance through data-driven approaches. However, without an appropriate upper bound for adversariality, the scenarios might exhibit excessive adversariality, potentially leading to unavoidable collisions. In this paper, we introduce *FREA*, a novel safety-critical scenarios generation method that incorporates the Largest *F*easible Region (LFR) of AV as guidance to ensure the *RE*asonableness of the *A*dversarial scenarios. Concretely, *FREA* initially pre-calculates the LFR of AV from offline datasets. Subsequently, it learns a reasonable adversarial policy that controls the scene's critical background vehicles (CBVs) to generate adversarial yet AV-feasible scenarios by maximizing a novel feasibility-dependent adversarial objective function. Extensive experiments illustrate that *FREA* can effectively generate safety-critical scenarios, yielding considerable near-miss events while ensuring AV's feasibility. Generalization analysis also confirms the robustness of *FREA* in AV testing across various surrogate AV methods and traffic environments. For more information, visit the project website: https://currychen77.github.io/FREA/

**Keywords:** Feasibility, Scenario Generation, Autonomous Driving

## 1 Introduction

Rapid advancements in AI-driven autonomous driving technologies have enabled autonomous vehicles (AVs) to operate efficiently, safely, and comfortably in most daily scenarios [1]. However, their performance in the complex real-world remains uncertain due to safety-critical yet rare scenarios, which are hard to collect at scale. Accurately generating these *safety-critical* scenarios and evaluating the AV performance is crucial for advancing autonomous driving systems [2].

Existing approaches primarily utilize metrics such as collision rate or minimum distance to AV to measure adversariality [3, 4, 5, 6]. These approaches aim to increase adversariality while complying with specific constraints. Methods based on Deep Generative Models (DGM) typically employ behavioral characteristics of traffic participants from natural datasets [6, 7] or apply domain-specific knowledge [8, 9, 10] to guide the generation of adversarial scenarios. Similarly, optimization-based methods [5, 11] incorporate additional regularization terms to maintain the rationality of agent behaviors. RL-based methods measure discrepancies between generated scenarios and macroscopic traffic distributions directly [12] or indirectly [3, 4] to ensure the naturalness of the adversarial scenarios. However, pursuing maximum adversariality in AV testing often leads to excessive adversarial scenarios. These scenarios typically result in unavoidable collisions, where no driving policy can ensure safety. This conflicts with our purpose of testing and improving the AV systems in scenarios

8th Conference on Robot Learning (CoRL 2024), Munich, Germany.

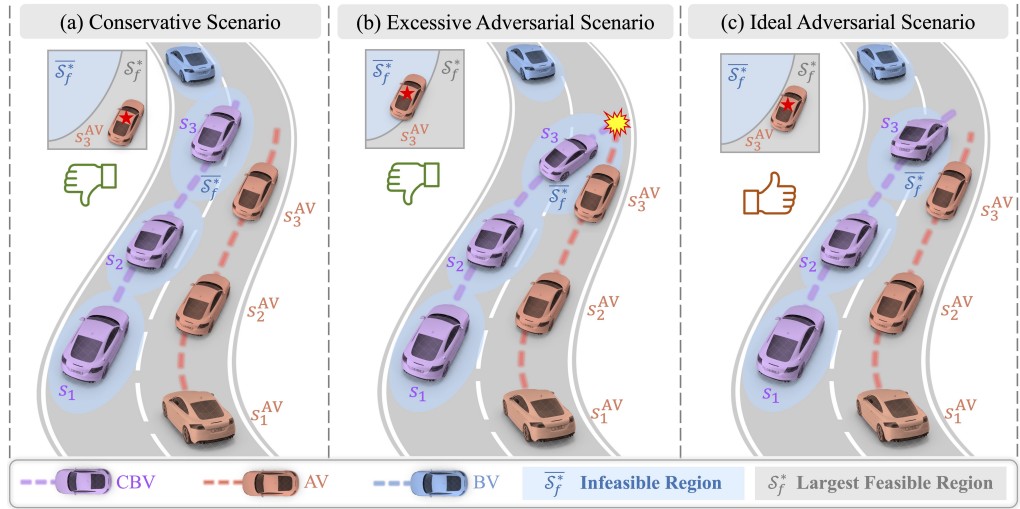

Figure 1: Illustration of adversarial yet AV-feasible scenarios in a two-lane traffic setting. The CBV employs three distinct policies, resulting in different scenarios: (a) Conservative scenario, where the policy is less adversarial; (b) Excessive adversarial scenario, resulting in an unavoidable collision; and (c) Ideal adversarial scenario, effectively balancing adversarial and AV's feasibility.

where collisions could have been avoided by taking proper actions. Therefore, the motivation of this work is to generate safety-critical scenarios with reasonable adversariality.

Inspired by the concept of *feasibility* in Safe Reinforcement Learning (Safe RL) [13, 14, 15], we propose an innovative approach using the feasibility of the AV to define the upper bound for adversariality in safety-critical scenarios. Specifically, under a given critical scenario, if an optimal policy exists that allows the AV to avoid potential collisions, then the current state of the AV is within its Largest Feasible Region (LFR), as detailed in Section 3.1. Therefore, we naturally use the boundary of the AV's LFR as the upper bound for the scenario's adversariality to ensure that even the most adversarial scenario generated still satisfies the AV's feasibility.

We illustrate several typical scenarios in Figure 1 to demonstrate adversarial yet AV-feasible scenarios intuitively. In Figure 1(a), the AV remains comfortably within its LFR, representing a conservative scenario. In contrast, Figure 1(b) depicts the AV entering the infeasible region, indicative of an excessive adversarial scenario that results in an unavoidable collision. Ideally, as demonstrated in Figure 1(c), the AV should remain within its LFR while closely approaching its boundary, thereby representing an adversarial yet AV-feasible scenario.

In this paper, we propose *FREA*, a novel method designed to generate adversarial yet AV-feasible scenarios. To address the "curse of dimensionality" [16], similar to approaches in [3, 4], we manipulate the behavior of some critical background vehicles (CBVs) at pivotal moments to reduce computational complexity. Moreover, we incorporate the AV's feasibility as guidance to ensure that the adversariality of CBV policy remains within a reasonable range. Specifically, *FREA* involves a two-stage framework: initially, we pre-train the optimal feasible value network of AV using offline datasets; subsequently, we introduce a feasibility-dependent adversarial objective that enables CBV to learn a reasonable adversarial policy. Experimental results demonstrate that feasibility-guided adversarial policy facilitates the generation of near-miss events while significantly reducing over-adversarial scenarios and unavoidable collisions. Further performance evaluation of AVs confirms the robustness and generalization ability of *FREA* in AV testing across various surrogate AV methods and environments. The main contributions of this work are concluded as follows:

1. We propose *FREA*, a safety-critical scenario generation method that leverages the Largest Feasible Region (LFR) of AV as guidance to ensure the adversarial reasonableness of the Critical Background Vehicles (CBVs), thereby generating adversarial yet AV-feasible scenarios.
2. We show that *FREA* facilitates generating near-miss events while ensuring AV's feasibility, and it also exhibits robustness in AV testing across various surrogate AV methods and environments.

## 2  Related Work

**Safety-Critical Scenarios Generation.** Recently, the Deep Generative Model (DGM) method has gained significant attention in scenario generation for its ability to construct neural networks that model data distributions and generate scenarios by sampling from these models [6, 7, 9, 10, 17]. Commonly, these methods output vehicle trajectories that may not always adhere to dynamic constraints or exhibit the necessary smoothness, raising concerns about their dynamic authenticity. In contrast, optimization-based methods [5, 11, 18] optimize the actions of all vehicles in a scene, ensuring that these actions stay within predefined constraints and maintain continuity to guarantee authenticity. These methods typically incorporate various regularization terms to enhance the rationality of the actions. However, with more vehicles, the "curse of dimensionality" [16] often hampers optimization-based methods from finding plausible trajectories, reducing performance [5, 11]. RL-based strategies [3, 4, 12, 19] address this by controlling only Critical Background Vehicles (CBVs) at crucial moments, aiming to enhance adversariality while ensuring consistency with natural dataset distributions. These methods typically evaluate adversariality using metrics like collision rate [3, 4], collision likelihood [19], or the minimum distance [12] to the AV. Although they emphasize macro-scenarios' naturalness, pursuing maximum adversariality still results in excessive adversarial scenarios. To mitigate these concerns, we introduce a feasibility-guided adversarial policy of CBV that ensures AV's feasibility while optimizing scenario adversariality.

**Safe RL with Hard Constraint.** Safe RL is usually modeled as a Constrained Markov Decision Process (CMDP) defined by $\mathcal{M} = (\mathcal{S}, \mathcal{A}, P, r, h, c, \gamma)$, where $\mathcal{S}$ and $\mathcal{A}$ represent the state and action spaces, $P$ describes the transition dynamics, $r : \mathcal{S} \times \mathcal{A} \to \mathbb{R}$ is the reward function, $h : \mathcal{S} \to \mathbb{R}$ is the constraint violation function, $c : \mathcal{S} \to [0, C_{\max}]$ is a bounded cost function, and $\gamma$ is the discount factor within $(0, 1)$. Typically, $c(s) = \max\{h(s), 0\}$. The goal is to optimize a policy $\pi$ to maximize rewards while adhering to safety constraints, especially worst-case constraints in safety-critical domains like autonomous driving. To enhance safety, various strategies have been developed: [20] proposes a safety index that ensures persistent safety for energy-dissipating policies, while [21, 13, 14] employ reachability certificates to ensure the worst-case constraints remain non-positive during online policy optimization. [15] decouples the learning of the reachability certificate from policy optimization, performing it offline for more stable training. As we need the LFR of the AV and online training with a fixed-policy AV only yields a subset of LFR, following [15], we adopt an offline method to comprehensively learn the AV's LFR, enhancing training stability.

## 3  Methodology

In this section, we integrate AV's feasibility into adversarial policy learning for the Critical Background Vehicle (CBV), formulating *FREA* as shown in Algorithm 1. The offline part (Section 3.1) trains the AV's optimal feasible value network using the offline dataset to approximate the Largest Feasible Region (LFR) of the AV. In the online part (Section 3.2), we propose a feasibility-dependent adversarial objective for CBV training with the predetermined LFR. For clarity, the elements associated with AV are denoted with $\cdot^{\mathrm{AV}}$, while CBV-related elements have no special superscripts.

### 3.1  Largest Feasible Region of AV

Given the efficacy of Hamilton-Jacobi Reachability (HJR) [22] in the domain of Safe RL, we adopt the HJR framework to instantiate LFR. First, we provide a brief overview of HJR (Definition 1) and then dive into the detailed instantiation of the LFR under the HJR framework (Definition 2).

**Definition 1** (Optimal feasible value function). *Based on [13, 15, 22], the optimal feasible state-value function $V_h^*$ and the optimal feasible action-value function $Q_h^*$ are defined in Eqs. (1) and (2).*

$$V_h^*(s^{\mathrm{AV}}) := \min_{\pi^{\mathrm{AV}}} \max_{t \in \mathbb{N}} h\left(s_t^{\mathrm{AV}}\right), s_0^{\mathrm{AV}} = s^{\mathrm{AV}}, a_t^{\mathrm{AV}} \sim \pi^{\mathrm{AV}}\left(\cdot \mid s_t^{\mathrm{AV}}\right), \tag{1}$$

$$Q_h^*(s^{\mathrm{AV}}, a^{\mathrm{AV}}) := \min_{\pi^{\mathrm{AV}}} \max_{t \in \mathbb{N}} h\left(s_t^{\mathrm{AV}}\right), s_0^{\mathrm{AV}} = s^{\mathrm{AV}}, a_0^{\mathrm{AV}} = a^{\mathrm{AV}}, a_{t+1}^{\mathrm{AV}} \sim \pi^{\mathrm{AV}}\left(\cdot \mid s_{t+1}^{\mathrm{AV}}\right). \tag{2}$$

The optimal feasible value functions exhibit the following properties:

- $V_h^*(s^{\text{AV}}) \leq 0 \Rightarrow \exists \pi^{\text{AV}}, \max_{t \in \mathbb{N}} h\left(s_t^{\text{AV}}\right) \leq 0$, that is, when $V_h^*(s^{\text{AV}})$ is not larger than zero, there exists an optimal policy for the AV that satisfies the hard constraints.
- $V_h^*(s^{\text{AV}}) > 0 \Rightarrow \forall \pi^{\text{AV}}, \max_{t \in \mathbb{N}} h\left(s_t^{\text{AV}}\right) > 0$, that is, when $V_h^*(s^{\text{AV}})$ is larger than zero, no policy can ensure the safety of AV under the hard constraints.

Therefore, the optimal feasible value function serves as an indicator of whether there exists a policy of AV that achieves hard constraints. Consequently, we can easily derive the following definition.

**Definition 2** (Largest Feasible Region (LFR)). *The largest feasible region is the sub-zero level set of the optimal feasible state-value function.*

$$\mathcal{S}_f^* := \left\{ s^{\text{AV}} \mid V_h^*(s^{\text{AV}}) \leq 0 \right\}$$

Although HJR effectively enforces hard constraints, deriving the optimal feasible value function typically requires Monte Carlo estimations through environmental interactions. However, in scenario generation, the AV usually adopts a fixed policy throughout these interactions. Consequently, the feasible region derived through interaction is merely a subset of LFR.

To accurately obtain the LFR, we adopt the method proposed in [15], which involves pre-collecting driving trajectories using different AV algorithms and then deriving the LFR through offline learning. The optimal feasible value functions $V_h^*(s^{\text{AV}})$ and $Q_h^*(s^{\text{AV}})$ can be obtained, respectively, by minimizing the following objective functions:

$$\mathcal{L}_{V_h}(\omega) = \mathbb{E}_{\mathcal{D}} \left[ L_{\text{rev}}^\tau \left( Q_h(s^{\text{AV}}, a^{\text{AV}}; \phi) - V_h(s^{\text{AV}}; \omega) \right) \right], \tag{3}$$

$$\mathcal{L}_{Q_h}(\phi) = \mathbb{E}_{\mathcal{D}} \left[ \left( \left( (1-\gamma)h(s^{\text{AV}}) + \gamma \max \left\{ h(s^{\text{AV}}), V_h(s^{\text{AV}'}; \omega) \right\} \right) - Q_h(s^{\text{AV}}, a^{\text{AV}}; \phi) \right)^2 \right], \tag{4}$$

where $L_{\text{rev}}^\tau(u) = |\tau - \mathbb{I}(u > 0)|u^2$ for $\tau \in (0.5, 1)$ and $\mathcal{D}$ is the offline dataset, comprising both safe and unsafe trajectories. Further dataset collection details are in Appendix C.1.

With the well-trained $V_h^*(s^{\text{AV}})$, we can easily assess whether the AV is within the LFR based on its current state $s^{\text{AV}}$. Additionally, by interpreting the AV's state as a scenario description and applying the viewpoint transformation function $g(\cdot)$, we derive $s^{\text{AV}}$ from $g(s)$, where $s$ is the CBV's state. This links the AV's LFR to the CBV's adversarial policy learning, showing $V_h^*(s^{\text{AV}}) = V_h^*(g(s))$, thus connecting the AV's feasibility with CBV policy.

## 3.2 Feasibility-Dependent Adversarial Objective of CBV

With the predetermined LFR of AV from Section 3.1 and the relation between $s^{\text{AV}}$ and $s$, the CBV can access AV's LFR through $s$ to evaluate feasibility. This enables us to form a standard CMDP framework for CBV to encourage adversarial behavior while ensuring reasonableness.

The standard feasibility-dependent objective function usually adopts distinct objects in the feasible and infeasible regions, with the principle of maximizing expected cumulative reward in the feasible region while minimizing constraint violations as much as possible in the infeasible region [13, 14, 15]. With the predetermined LFR and the data $(s, a, s', r)$ sampled from the rollout buffer, we modified the on-policy version of the feasibility-dependent objective function from [13], incorporating stricter constraints, as shown in Eqs. (5) and (6). Intuitively, we hope to maximize CBV's adversarial rewards when the current state $s^{\text{AV}} = g(s)$ and the next state $s^{\text{AV}'} = g(s')$ of the AV are within LFR and minimize the AV's constraint violations otherwise (Appendix A.1 demonstrates the theoretical relationships between Eq. (6) and the origin advantage function in [13]).

$$L(\theta) = \mathbb{E}_{\pi_{\theta_k}} \left[ \min \left( r_t(\theta) A^{\pi_{\theta_k}}(s, a), \text{clip} \left( r_t(\theta), 1 - \epsilon, 1 + \epsilon \right) A^{\pi_{\theta_k}}(s, a) \right) \right], \tag{5}$$

$$A^{\pi_{\theta_k}}(s, a) = A_r^{\pi_{\theta_k}}(s, a) \cdot I(s, s') + A_h^*(s^{\text{AV}}, a^{\text{AV}}) \cdot (1 - I(s, s')), \tag{6}$$

where $r_t(\theta) = \frac{\pi_\theta(a|s)}{\pi_{\theta_k}(a|s)}$ is the ratio, $A_r^{\pi_{\theta_k}}(s, a) = Q_r^{\pi_{\theta_k}}(s, a) - V_r^{\pi_{\theta_k}}(s)$ is CBV's reward advantage, $A_h^*(s^{\text{AV}}, a^{\text{AV}})$ is AV's feasibility advantage, and $I(s, s') = \mathbb{I}_{V_h^*(g(s)) \leq 0, V_h^*(g(s')) \leq 0}$ is the indicator.

---

**Algorithm 1** Feasibility-guided reasonable adversarial policy (*FREA*)

---

1: **Offline Part** (Section 3.1)
2: Initialize feasibility value networks $V_h, Q_h$.
3: **for** each gradient step **do**
4:     Update $V_h$ using Eq. (3)    # Optimal feasible state-value function learning
5:     Update $Q_h$ using Eq. (4)    # Optimal feasible action-value function learning
6: **end for**
7: **Online Part** (Section 3.2)
8: Initialize policy parameters $\theta_0$, reward value function parameters $\psi_0$
9: **for** $k = 0, 1, 2, \dots$ **do**
10:     Collect set of trajectories $\mathcal{B}_k = \{\tau_i\}$ with policy $\pi_{\theta_k}$, where $\tau_i$ is a $T$-step episode.
11:     Compute reward advantage $A_r^{\pi_{\theta_k}}(s, a)$, using generalized advantage estimator (GAE [23]).
12:     Compute feasibility advantage using Eq. (9).
13:     Derive overall advantage using Eq. (6)    # Advantage calculating
14:     Fit reward value function, by Smooth L1 Loss.    # Value function learning
15:     Update the policy parameters $\theta$ by maximizing Eq. (5).    # Policy learning
16: **end for**

---

According to Eq. (2), $Q_h^*(s^{\mathrm{AV}}, a^{\mathrm{AV}})$ depends exclusively on $h(\cdot)$, which is in turn influenced by $s_t^{\mathrm{AV}}, t \in \mathbb{N}$. The action $a^{\mathrm{AV}}$ affects only the next state $s^{\mathrm{AV}'}$ through the environment transition function. Furthermore, when the BVs follow a deterministic policy, we can reasonably assume that the environment, excluding the CBV and AV, is deterministic. Therefore, we can express the next state of the AV as $s^{\mathrm{AV}'} = P\left(s^{\mathrm{AV}}, a^{\mathrm{AV}}, s, a\right)$, where $P$ represents the deterministic environment transition function. This framework allows for the assertion that given the data sample, $\left(s^{\mathrm{AV}}, a^{\mathrm{AV}}, s, a\right)$, the next state of the AV is thereby determined, allowing us to derive the following lemma:

**Lemma 1.** *As the BVs follow deterministic policy, the optimal feasible action-value function of AV can be achieved by AV's current state and next state (see Appendix A.2 for proof).*

$$Q_h^*\left(s^{\mathrm{AV}}, a^{\mathrm{AV}}\right) = \begin{cases} V_h^*(s^{\mathrm{AV}'}) & h(s^{\mathrm{AV}'}) \geq h(s^{\mathrm{AV}}) \\ \max\{h(s^{\mathrm{AV}}), V_h^*(s^{\mathrm{AV}'})\} & h(s^{\mathrm{AV}'}) < h(s^{\mathrm{AV}}) \end{cases} \tag{7}$$

With Lemma 1 and the viewpoint transformation function $s^{\mathrm{AV}} = g(s)$, we can derive AV's feasibility advantage through CBV's current state $s$ and next state $s'$ as:

$$A_h^*(s^{\mathrm{AV}}, a^{\mathrm{AV}}) = Q_h^*\left(s^{\mathrm{AV}}, a^{\mathrm{AV}}\right) - V_h^*\left(s^{\mathrm{AV}}\right) \tag{8}$$

$$= \begin{cases} V_h^*\left(g\left(s'\right)\right) - V_h^*\left(g\left(s\right)\right) & h\left(g\left(s'\right)\right) \geq h\left(g\left(s\right)\right) \\ \max\{h\left(g\left(s\right)\right), V_h^*\left(g\left(s'\right)\right)\} - V_h^*\left(g\left(s\right)\right) & h\left(g\left(s'\right)\right) < h\left(g\left(s\right)\right) \end{cases} \tag{9}$$

By integrating Eqs. (5), (6) and (9), we formulate the online part of *FREA* in Algorithm 1.

## 4 Experiments

Following the experimental setup (Section 4.1), we conducted several experiments to affirm: The Largest Feasible Region (LFR) of the AV is reasonable through offline training (Section 4.2); *FREA* effectively generates near-miss events while maintaining AV's feasibility (Section 4.3); *FREA* shows generalization in AV testing across various surrogate AVs and environments (Section 4.4); and *FREA* continuously creates adversarial yet AV-feasible scenarios in complex traffic (Section 4.5).

### 4.1 Experiment Setup

**Environment.** We developed our environment using the Carla Simulator [24] in accordance with SafeBench [25], and combined "Scenario6" and "Scenario7" into "Scenario9", spanning across "Town05" and "Town02". In "Scenario9", the AV navigates predefined routes through various junctions, avoiding randomly placed background vehicles (BVs). To implement reasonable adversarial

attacks, we applied the *FREA* method to Critical Background Vehicles (CBVs), attacking the surrogate AV with a goal-based adversarial reward to create adversarial yet AV-feasible, safety-critical scenarios. More experimental details are provided in Appendix B.

**Baselines.** We evaluate **FREA** against three other CBV methods as baselines: (1) **Standard:** a rule-based autopilot policy from Carla; (2) **PPO:** a PPO-based adversarial policy without feasibility guidance; (3) **FPPO-RS:** a PPO-based adversarial policy that includes a fixed penalty to the reward whenever the AV enters the infeasible region. (Further algorithm details are in Appendix B.3)

## 4.2 Largest Feasible Region Learning of AV

According to Eqs. (3) and (4), learning the optimal feasible value functions incorporates the constraint violation function $h(s^{AV})$. A simple definition of $h(s^{AV})$ is $\min_{i \in \{1,...,N\}} d(AV, BV^i)$, where $d(AV, BV^i)$ represents the minimum distance between the bounding boxes of AV and $BV^i$. However, since the minimum distance is non-negative, we adopt a sparse design for $h(s^{AV})$, which helps to maintain a stable training balance of positive and negative samples, as advised by [15]:

$$h(s^{AV}) := \begin{cases} -1 & \text{if } \min_{i \in \{1,...,N\}} d(AV, BV^i) > d_{th} \\ M & \text{if } \min_{i \in \{1,...,N\}} d(AV, BV^i) \leq d_{th} \end{cases}, \qquad (10)$$

where $N$ is the total number of BVs and $d_{th}$ and $M$ are safety-related hyperparameters. Further training and application details about the LFR are provided in Appendix C.

Figure 2 depicts the well-trained LFR of the AV in varying traffic scenarios. Figure 2(a) and (b) reveal that the infeasible region expands as the AV's speed increases, indicating higher collision risks. Figure 2(c) and (d) confirm the LFR's consistency across scenarios. These cases show the pre-trained LFR's reliability under various scenarios, providing a solid basis for CBV policy training.

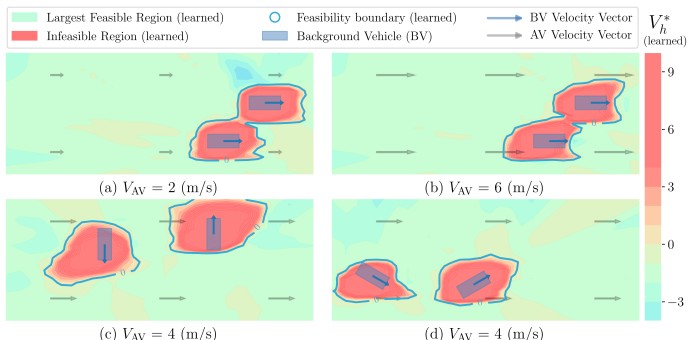

Figure 2: Visualization of well-trained LFR in various scenarios.

## 4.3 Quantitative Analysis of Generated Scenarios

To verify that *FREA* can generate adversarial yet AV feasible scenarios, we quantified the near-miss events and the AV's feasibility across various CBV methods. To ensure consistency and fairness in our quantitative analysis, we employed the same surrogate AV method, "Expert" [26], throughout the experiments. Further analysis about the surrogate AV methods is discussed in Section 4.4.

**Evaluation of Near-miss Events.** Previous research indicates that all safety-related incidents share common origins with near-misses [27, 28, 29], highlighting the importance of analyzing near-miss events in the generation of safety-critical scenarios. Consistent with [4], we use time-to-collision (TTC) in [30] and post-encroachment time (PET) in [4] as metrics to quantify these near-miss events.

Figure 3(a) and (b) show the distributions of TTC and PET across different CBV methods. The PPO produces the most adversarial scenarios across all maps. Conversely, FPPO-RS and *FREA* exhibit similar near-miss results but with a higher frequency of near-miss than Standard traffic flow. Given our focus on AV-feasible near-miss events, further analysis of AV's feasibility is required.

**Evaluation of AV's Feasibility.** While near-miss events are crucial for generating safety-critical scenarios, it is also vital to avoid over-collision events caused by excessive adversarial CBV policy. To ensure that *FREA* effectively minimizes over-collision events, we analyze the scenarios under different CBV methods with four metrics: (1) Collision Rate (CR); (2) Infeasible Ratio (IR); (3)

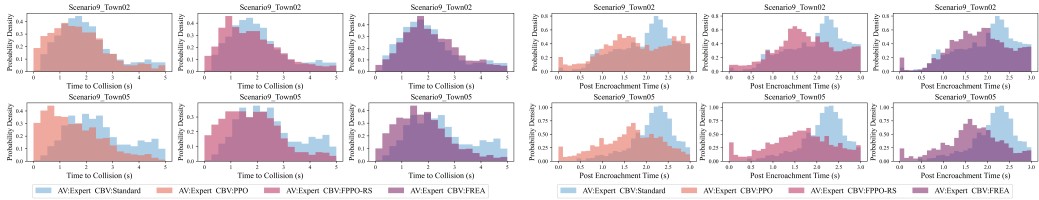

(a) Time to collision distribution                    (b) Post encroachment time distribution

Figure 3: Evaluating near-miss events across different CBV methods with Expert [26] as AV

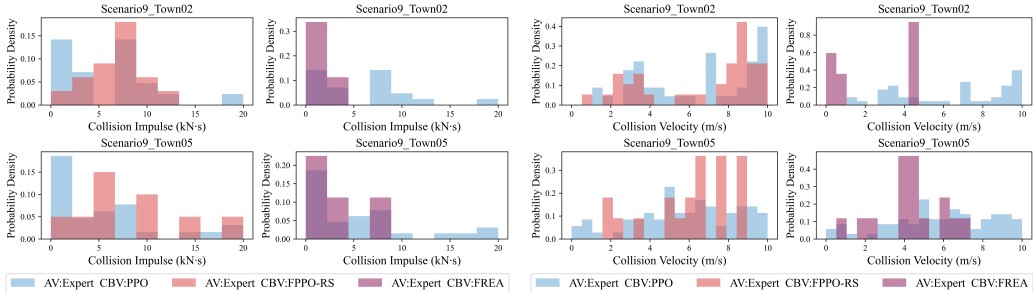

(a) Collision impulse distribution                    (b) Collision velocity distribution

Figure 4: Evaluating collision severity across different CBV methods with Expert [26] as AV

Table 1: Feasibility evaluation using Expert [26] as AV under different CBV methods. Results are the average of 10 runs in "Scenario9" with varied seeds.

| CBV | Feasibility | Town05 intersections | | | Town02 intersections | | |
|---|---|---|---|---|---|---|---|
| | | CR (↓) | IR (↓) | ID (↓) | CR (↓) | IR (↓) | ID (↓) |
| KING[5] | ✗ | 76.67% | 65.97% | 7.54m | N/A | N/A | N/A |
| PPO | ✗ | 37.5% | 35.56% | 10.57m | 30.0% | 51.18% | 12.40m |
| FPPO-RS | ✓ | 11.25% | 34.92% | 9.13m | 24.29% | 45.36% | 9.16m |
| **FREA** | ✓ | **5.0%** | **31.10%** | **6.25m** | **5.71%** | **27.18%** | **4.94m** |

Infeasible Distance (ID); and (4) Collision Severity, including collision speed and impulse. See Appendix E.2 for metric details and Appendix B.3 for detailed reimplementation of KING [5].

As shown in Figure 4 and Table 1, without the feasibility constraint, the PPO and KING [5] lead to many severe collisions, indicative of excessive adversarial behavior. In contrast, under the feasibility constraint, both FPPO-RS and *FREA* significantly reduce the severity of these events. However, FPPO-RS still struggles with hard constraints, leading to many infeasible events. Conversely, *FREA* balances adversarial intensity with AV's feasibility, thereby achieving the least severe collisions.

In conclusion, our analysis confirms that *FREA* effectively generates considerable near-miss events while ensuring AV's feasibility, thus creating adversarial yet AV-feasible scenarios.

### 4.4 Generalization Analysis of AV Testing.

To evaluate the generalization of *FREA* across various surrogate AVs and environments in AV testing, we assessed the driving performance of PlanT [31] and Expert [26] under different CBVs that were pre-trained with specific surrogate AVs on "Town05" and "Town02". The performance metrics follow the setting in [25], where the OS represents the overall score (details in Appendix E.3).

Table 2 illustrates that the PPO method, pre-trained with various surrogate AV methods, exhibits notable variance in AV testing. This variance suggests that, without feasibility constraints, the CBV's adversarial policy closely depends on the specific surrogate AV method. In contrast, *FREA* demonstrates remarkable stability in AV testing. This stability is primarily due to goal-based adversarial rewards and feasibility constraints, both independent of the specific surrogate AV method employed. Furthermore, *FREA* consistently aligns its relative AV testing results with standard traffic flows, suggesting that *FREA* facilitates safety-critical scenario generation while ensuring unbiased AV testing.

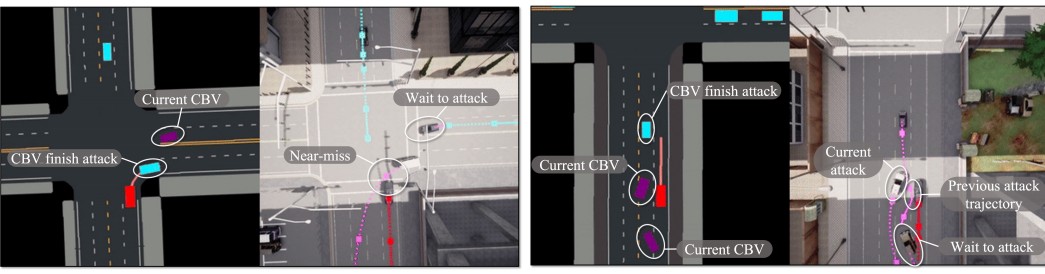

(a) Intersection attacks              (b) Straight road attacks

Figure 5: Representative scenarios: Red: AV (Expert [26]). Blue: BV. Purple: CBV (*FREA*).

Table 2: Comparative performance of AVs across different maps, using CBV methods pre-trained with various surrogate AVs. Results are the average of 10 runs in "Scenario9" with varied seeds.

| CBV | Surr. AV | AV | Town05 intersections | | | | | | Town02 intersections | | | | | |
|---|---|---|---|---|---|---|---|---|---|---|---|---|---|---|
| | | | CR (↓) | OR (↓) | RF (↓) | UC (↓) | TS (↓) | OS (↑) | CR (↓) | OR (↓) | RF (↓) | UC (↓) | TS (↓) | OS (↑) |
| Standard | ✗ | Expert | 0.0% | 0.0m | 7.0m | 1% | 55s | 94.0 | 0.0% | 0.0m | 6.0m | 2% | 63s | 93.0 |
| | | PlanT | 1.0% | 0.0m | 7.0m | 6% | 70s | 90.0 | 1.0% | 0.0m | 6.0m | 6% | 76s | 90.0 |
| PPO | Expert | Expert | 36.0% | 0.0m | 6.0m | 8% | 66s | 76.0 | 40.0% | 0.0m | 6.0m | 15% | 66s | 72.0 |
| | | PlanT | 61.0% | 1.0m | 7.0m | 11% | 70s | 65.0 | 70.0% | 0.0m | 6.0m | 27% | 64s | 57.0 |
| PPO | PlanT | Expert | 26.0% | 0.0m | 6.0m | 8% | 64s | 80.0 | 21.0% | 0.0m | 6.0m | 12% | 74s | 80.0 |
| | | PlanT | 45.0% | 0.0m | 7.0m | 7% | 69s | 72.0 | 51.0% | 0.0m | 6.0m | 18% | 70s | 67.0 |
| **FREA** | Expert | Expert | 4.0% | 0.0m | 7.0m | 7% | 67s | **89.0** | 9.0% | 0.0m | 6.0m | 16% | 75s | **83.0** |
| | | PlanT | 10.0% | 0.0m | 7.0m | 5% | 73s | **86.0** | 10.0% | 0.0m | 7.0m | 24% | 86s | **79.0** |
| **FREA** | PlanT | Expert | 5.0% | 0.0m | 7.0m | 5% | 62s | **90.0** | 14.0% | 0.0m | 6.0m | 15% | 75s | **82.0** |
| | | PlanT | 9.0% | 0.0m | 7.0m | 6% | 73s | **87.0** | 17.0% | 0.0m | 7.0m | 18% | 83s | **79.0** |

## 4.5 Analysis of Representative Scenarios

To confirm the validity of scenarios generated by *FREA*, we analyze representative cases. Figure 5(a) illustrates a scenario at an intersection where the AV, attempting a right turn, is cut off by a faster CBV also turning right prematurely, forcing the AV to stop and wait. Subsequently, another vehicle to the right of the AV is selected as the next CBV for further attack. Figure 5(b) depicts a scenario on a straight road where multiple CBVs sequentially adopt different strategies to attack the AV. These examples demonstrate that *FREA* is capable of creating a series of continuous, adversarial yet AV-feasible scenarios in complex traffic environments. (Further analysis can be found in Appendix F)

## 5 Conclusion

In conclusion, *FREA* employs the concept of the Largest Feasible Region (LFR) to establish the upper bound of adversariality in safety-critical scenario generation. This approach aims to maximize adversariality while ensuring AV's feasibility. Specifically, *FREA* employs a two-stage framework that begins by pre-training the AV's optimal feasible value network using offline datasets to establish the LFR. Then, it learns a reasonable adversarial policy for Critical Background Vehicles (CBVs) based on a feasibility-dependent adversarial objective, thereby creating adversarial yet AV-feasible scenarios. Experimental results illustrate that the scenarios generated by *FREA* contain considerable near-miss events while effectively reducing unavoidable collision events. Moreover, guided by the AV's feasibility and the goal-based adversarial reward, *FREA* demonstrates enhanced robustness and generalization ability in AV testing across various environments and surrogate AV methods.

**Limitation.** With all the advantages of *FREA* discussed, some limitations of our method also suggest directions for future work. As detailed in Appendix C.1, the LFR of AV exhibits instability under different hyperparameters. This instability is primarily due to the limited offline data size, a concern also highlighted in [15], which adversely affects the accuracy of LFR assessments. In addition, *FREA* lacks certain traffic common sense and is prone to violating traffic regulations, as discussed in Appendix F.2. In the future, integrating intelligent agents like LLMs to address these deficiencies and ensure compliance with traffic rules will be an important research direction.

**Acknowledgments**

This research was supported by the National Natural Science Foundation of China under Grant 52221005 and the Tsinghua-Toyota Joint Research Institute Interdisciplinary Project under Grant 20223930039.

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

# A  Theoretical Analysis

## A.1  Feasibility-depend Advantage Function

In [13], the feasibility-dependent advantage function relates to $A_r^{\pi_{\theta_k}}(s, a)$ and $A_h^{\pi_{\theta_k}}(s, a)$. However, as we use the Largest Feasible Region (LFR) of the AV as the guidance to train the CBV policy and the predetermined LFR is independent of the CBV policy $\pi$, $A_h^{\pi_{\theta_k}}(s, a)$ can be substituted with $A_h^*(s^{AV}, a^{AV})$. Thus, the combined advantage function is defined as:

$$A^{\pi_{\theta_k}}(s, a) = A_r^{\pi_{\theta_k}}(s, a) + \lambda_\xi(s) \cdot A_h^*(s^{AV}, a^{AV}), \tag{11}$$

where $\lambda_\xi(s)$ acts as an indicator function. For feasible states, $\lambda_\xi(s) \to 0$ is finite, and for unfeasible states, $\lambda_\xi(s) \to +\infty$. Consequently, the feasibility-dependent advantage function simplifies to:

$$A^{\pi_{\theta_k}}(s, a) = A_r^{\pi_{\theta_k}}(s, a) \cdot \mathbb{I}_{s \in \mathcal{S}_f^*} + A_h^*(s^{AV}, a^{AV}) \cdot \mathbb{I}_{s \notin \mathcal{S}_f^*} \tag{12}$$

Aligning with Definition 2 and the viewpoint transformation function $g(\cdot)$, we reinterpret $\mathcal{S}_f^*$ using the optimal feasible state-value function $V_h^*$, which is relative to the AV's state:

$$A^{\pi_{\theta_k}}(s, a) = A_r^{\pi_{\theta_k}}(s, a) \cdot \mathbb{I}_{V_h^*(g(s)) \leq 0} + A_h^*(s^{AV}, a^{AV}) \cdot \mathbb{I}_{V_h^*(g(s)) > 0} \tag{13}$$

Given that the PPO-based method calculates the advantage solely from the trajectories stored in the buffer, we propose imposing stricter constraints on $A_r^{\pi_{\theta_k}}(s, a)$. Specifically, if the AV's next state falls outside the LFR, the optimization should prioritize minimizing feasibility violations over maximizing adversarial rewards. Consequently, we formulate the final advantage function in Eq. (6).

## A.2  Proof of Lemma 1

Given that other BVs adhere to a consistent rule-based policy, we can reasonably assume that the environment, excluding the CBV and AV, is deterministic. Thus, we can redefine Eq. (2) as follows:

$$Q_h^*(s^{AV}, a^{AV}) := \min_{\pi^{AV}} \max_{t \in \mathbb{N}} \left\{ h\left(s_0^{AV}\right), h\left(s_{t+1}^{AV}\right) \right\},$$
$$s_0^{AV} = s^{AV}, a_0^{AV} = a^{AV}, s_0 = s, a_0 = a, s_{t+1}^{AV} = P\left(s_t^{AV}, a_t^{AV}, s_t, a_t\right), s_{t+1} = g\left(s_{t+1}^{AV}\right)$$
$$a_{t+1}^{AV} \sim \pi^{AV}\left(\cdot \mid s_{t+1}^{AV}\right), a_{t+1} \sim \pi^{CBV}\left(\cdot \mid s_{t+1}\right), \tag{14}$$

where $s$ and $a$ are the state-action pair of the CBV and $P$ is the deterministic transition function excluding the CBV and AV.

Additionally, based on Definition 1 and Eq. (14), we identify two cases:

Case1:  if $h(s^{AV'}) \geq h(s^{AV})$, then:

$$Q_h^*(s^{AV}, a^{AV}) = \min_{\pi^{AV}} \max_{t \in \mathbb{N}} \left\{ h\left(s_0^{AV}\right), h\left(s_{t+1}^{AV}\right) \right\} = \min_{\pi^{AV}} \max_{t \in \mathbb{N}} \left\{ h\left(s_{t+1}^{AV}\right) \right\} = V_h^*(s^{AV'}) \tag{15}$$

Case2:  if $h(s^{AV'}) < h(s^{AV})$, then:

$$Q_h^*(s^{AV}, a^{AV}) = \min_{\pi^{AV}} \max_{t \in \mathbb{N}} \left\{ h\left(s_0^{AV}\right), h\left(s_{t+1}^{AV}\right) \right\} = \max\{h(s^{AV}), V_h^*(s^{AV'})\} \tag{16}$$

Building upon Eqs. (15) and (16), we conclude in Eq. (17) that $Q_h^*(s^{AV}, a^{AV})$ primarily depends on the states of the AV. The actions of both the AV and the CBV mainly influence state transitions.

$$Q_h^*(s^{AV}, a^{AV}) = \begin{cases} V_h^*(s^{AV'}) & h(s^{AV'}) \geq h(s^{AV}) \\ \max\{h(s^{AV}), V_h^*(s^{AV'})\} & h(s^{AV'}) < h(s^{AV}) \end{cases} \tag{17}$$

# B Experiment Details

Building on foundational concepts from [3, 4], we apply our *FREA* method to critical background vehicles (CBVs) within traffic flows. We first outline the mechanisms for specifying and withdrawing CBVs, as detailed in Appendix B.1. We then discuss the safe RL-based setting implemented in our *FREA* method, described in Appendix B.2. Finally, we provide implementation details of the baseline methods in Appendix B.3.

## B.1 Specifying and Withdrawal of CBVs

To appropriately select CBVs and exclude unsuitable BVs, we established criteria to filter out ineligible candidates:

Case1: The BV is located in the opposing lane relative to the AV.

Case2: The distance between the BV and AV exceeds 25 meters.

Case3: The BV is positioned behind the AV with a relative yaw angle greater than 90 degrees, indicating no interaction.

Case4: The BV has previously served as a CBV and has reached its goal in the scenario.

Based on the predefined criteria, our system assesses the scenario at each simulation step in Carla. If the number of active CBVs drops below a predefined threshold, the nearest candidate to the AV is automatically selected as a new CBV. To prevent disruptions in normal traffic flow during training, we implemented a withdrawal mechanism for CBVs that specifies conditions for their removal, whether they complete their tasks or not.

Case1: The CBV achieves its objective (it is then terminated and reverts to a standard BV).

Case2: The BV is positioned behind the AV with a relative yaw angle greater than 90 degrees (it is truncated and reverts to a standard BV).

Case3: The CBV obstructs traffic flow or exceeds the maximum allowed duration (it is truncated and reverts to a standard BV).

Case4: The CBV collides with any BV or the AV (it is terminated and removed from the simulation).

This selection and withdrawal mechanism effectively manages the CBV training process. However, considering the potential limitations of these rules, developing more intelligent strategies remains a future research direction.

## B.2 Safe RL-based Setting

In this framework, each CBV acts as an RL agent tasked with attacking the AV while maintaining the AV's feasibility. Previous works [5, 12] primarily focused on minimizing the distance between CBV and AV, often leading to unavoidable collisions and unrealistic scenarios, as discussed in Section 2. To mitigate this, we replace the collision reward with the goal-based adversarial reward, which encourages the CBV to reach a potential conflict point with the AV. The configuration details are as follows:

**State.** The state of each CBV is represented by an array of dimensions $(V + 2) \times F$, where $V$ represents the number of nearby vehicles and $F$ the features of these vehicles, the AV, and the goal point. Recorded features include relative x and y positions, object extent along the X and Y axes, relative yaw angle, and absolute speed. For the goal point, speed is replaced by relative distance. The features of the AV and the goal point are positioned in the first and second rows of the array, respectively.

**Action.** Following the guidelines in [25], we define a continuous action space with specific constraints to prevent unreasonable actions. The acceleration range is set between $-3$ and $3$, and the steering angle is limited to a maximum absolute value of $0.3$.

**Reward.** As previously discussed, we have replaced the collision reward with a goal-based reward. This adjustment encourages the CBV to navigate toward a potential conflict point while avoiding collisions with other BVs. In practice, we set the reference point in the global route of the AV as the potential conflict point, which forms the overall reward function:

$$R_t = d\left(\text{CBV}_{t-1}, \text{Goal}_{t-1}\right) - d\left(\text{CBV}_t, \text{Goal}_t\right) + 15 * r_t^{collision} + 15 * r_t^{finish}, \qquad (18)$$

where $d\left(\text{CBV}_t, \text{Goal}_t\right)$ denotes the Euclidean distance between the CBV and its goal point at time $t$, the collision reward $r_t^{collision}$ is set to $-1$ if the CBV collides with any BVs at time $t$, and the goal-reaching reward $r_t^{finish}$ is set to 1 if the CBV is within 2 meters radius of the goal at time $t$.

### B.3 Implementation Details of Baselines.

### B.3.1 Algorithms.

To evaluate the performance of *FREA*, we propose three CBV methods as baselines for a comprehensive quantitative comparison:

**Standard.** This method utilizes a rule-based autopilot policy implemented in the Carla Simulator [24] to generate realistic urban traffic flows.

**PPO.** This method employs an adversarial CBV policy based on PPO that aims to reach a potential conflict point with the AV, utilizing the adversarial reward outlined in Eq. (18).

**FPPO-RS.** This method integrates a feasibility penalty term into the PPO-based adversarial policy, which penalizes violations of the AV's feasibility constraints in the adversarial reward function. The modified adversarial reward function used during training is specified as follows:

$$R_t^{fea} = R_t - \frac{\text{clip}\left(V_h^*\left(g(s_t)\right), 0, f_{max}\right) \cdot p_{max}}{f_{max}}, \qquad (19)$$

where $f_{max}$ represents the upper bound for feasible value clipping, and $p_{max}$ is the upper bound for penalty rewards.

**KING.** This method reimplements the optimization-based safety-critical scenario generation approach, KING, as presented in [5]. Given the differences between our general experimental setup and that of KING, we made several necessary adjustments. For example, KING's experimental setup includes background vehicle counts limited to 1, 2, and 4, while we employ a continuous traffic flow with more than 10 vehicles. Additionally, the target speed for the AV in KING is set at $4\ m/s$, whereas we use a typical speed of $6\ m/s$. KING's experiments are conducted across Town03 to Town06, while our work focuses on Town05 and Town02. Therefore, KING is not directly comparable with the mentioned baselines in experiments that quantify near-miss events and assess the generality of AV testing. As a result, we limit our comparison of KING with the baselines to AV's feasibility experiments, which are less influenced by variations in traffic flow settings. Specifically, we adjust the target speed of the AV in KING from $4\ m/s$ to $6\ m/s$ and restrict our comparative analysis to results from Town05 to ensure fairness.

### B.3.2 Hyperparameters.

Table 3 shows the hyperparameters of baseline methods.

**Training Curves about CBV methods.** To ensure robustness in our training process, we aggregated results from three different random seeds, as shown in Figure 6. This illustration confirms that all three CBV methods converge well in various scenarios. Furthermore, the PPO method achieves the highest adversarial reward, reflecting its focus on adversarial objectives. Conversely, FPPO-RS and *FREA* need to balance adversariality and AV's feasibility, resulting in a lower adversarial reward. Given our focus on creating reasonable adversarial scenarios, achieving appropriate adversariality is more important than merely pursuing high adversarial rewards.

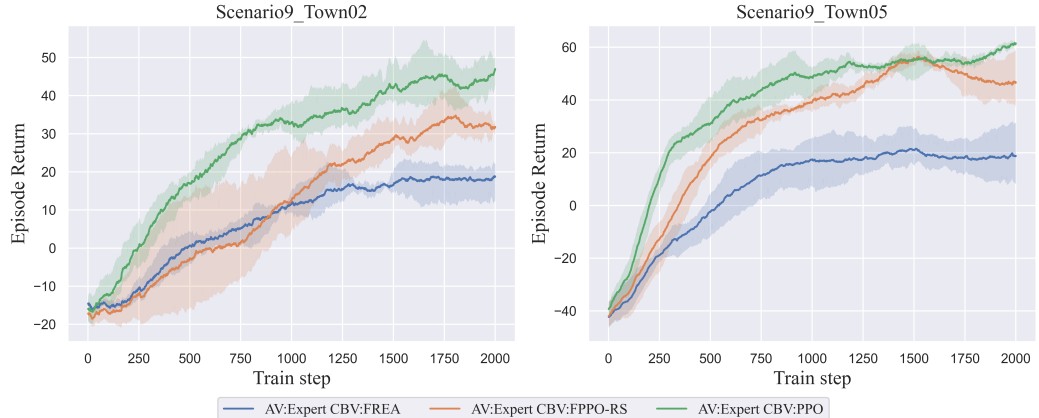

Figure 6: Episode return of different CBV methods.

Table 3: Detailed hyperparameters of *FREA* and baselines.

| Parameter | Value |
|---|---|
| **PPO, FPPO-RS, *FREA* shared.** | |
| Optimizer | Adam ($\epsilon = 1e-5$) |
| Approximation function | Multi-layer Perceptron |
| Number of hidden layers | 2 |
| Number of hidden units per layer | 256 |
| Nonlinearity of hidden layer | RELU |
| Nonlinearity of output layer | linear |
| Critic learning rate | Linear annealing $3e-4 \to 0$ |
| Reward discount factor ($\gamma$) | 0.98 |
| GAE parameters ($\lambda$) | 0.98 |
| entropy parameters | 0.01 |
| Batch size | 256 |
| horizon length | 2048 |
| update repeat times: | 4 |
| Max episode length ($N$) | 2000 |
| Actor learning rate | Linear annealing $3e-4 \to 0$ |
| Clip ratio | 0.2 |
| **FPPO-RS** | |
| feasible value clip upper bound $f_{max}$ | 8 |
| penalty reward upper bound $p_{max}$ | 1 |

## C  Largest Feasible Region: Training and Application

### C.1  Training Details about LFR

**Offline Datasets.** As highlighted in [15], extensive coverage of the state space in datasets is crucial for determining the LFR of AVs using offline RL. Following this guideline, we employed the Expert [26] and Behavior [24] agents as surrogate AVs, collecting $100k$ instances of interaction data under standard traffic conditions for each. Additionally, to introduce randomness, we employed the PPO method as CBV with the Expert as AV, gathering another $100k$ instances of interaction data. This resulted in a comprehensive dataset of $300k$ interaction data for LFR training. As depicted in Figure 7, the offline dataset extensively covers most of the potential state space, satisfying the requirements for training an optimal feasible value function.

**Constrain Function Setting.** As outlined in Eq. (10), the hyperparameters $M$ and $d_{th}$, along with the minimum distance between vehicles' bounding boxes, are crucial for defining $h\left(s^{AV}\right)$. Specifi-

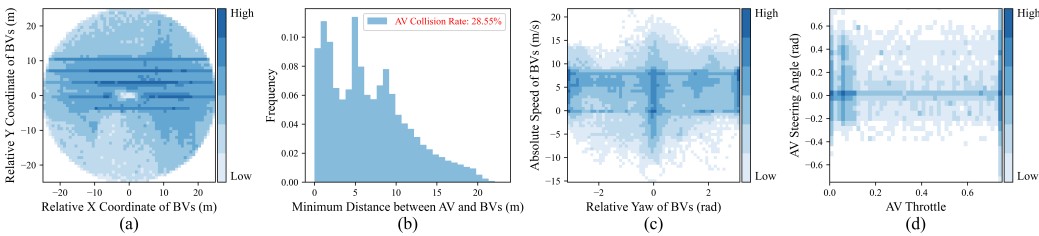

Figure 7: The offline data distribution.

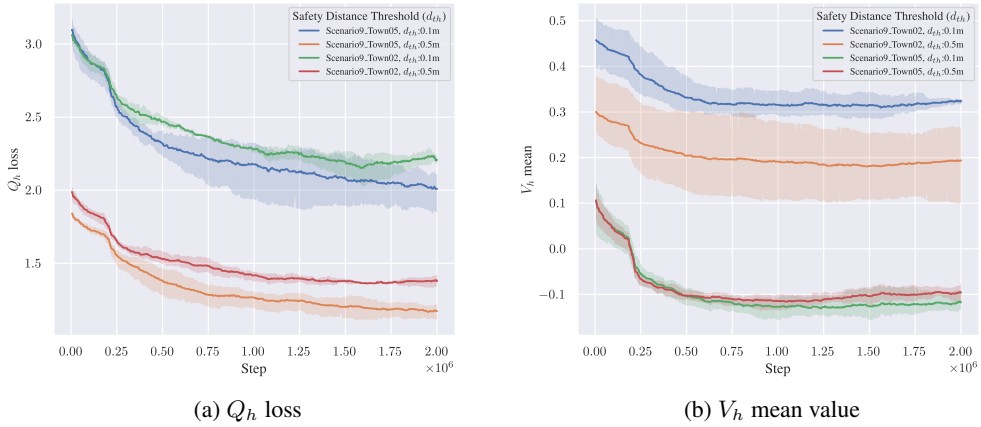

(a) $Q_h$ loss
(b) $V_h$ mean value

Figure 8: Learning curves in LFR training

cally, we employed the method described in [32] to calculate the minimum distance, which is always non-negative. To ensure a balance between positive and negative samples, we experimented with various settings for $M$ and $d_{th}$.

In our experiment, we set $d_{th}$ at 0.1 meters and 0.5 meters in separate trials to find the optimal parameter combination. Drawing on the findings from [15], we determined that optimal performance is achieved when the mean of the feasible value function is close to zero. Through empirical testing, we established $d_{th} = 0.1m, M = 18$ and $d_{th} = 0.5m, M = 12$ as the optimal settings. Given that these parameters are used to train across multiple towns, it is crucial to balance their optimality for various environments. Figure 8 illustrates the learning curves for these settings, confirming that the chosen parameters effectively maintain the mean values of the feasible value function in different towns close to zero.

The visualization results for $d_{th} = 0.1m, M = 18$ are presented in Figure 2, while those for $d_{th} = 0.5m, M = 12$ are shown in Figure 9. The former parameter set proved to be more effective in identical scenarios, leading us to select $d_{th} = 0.1m, M = 18$ for subsequent CBV training.

Detailed hyperparameter settings for the feasible value function can be found in Table 4.

### C.2 Application Details of LFR

During the feasibility-guided adversarial policy training for CBV, we encountered a significant challenge: the optimal feasible value function produces a single scalar value at each timestep. In scenarios with multiple CBVs, it is difficult to identify which CBV renders the AV's operation unfeasible. To address this issue, we introduced a "pseudo" state for each CBV at every timestep. This "pseudo" state captures information relevant to both the AV and CBV from the perspective of the AV. Crucially, this "pseudo" state is utilized solely to assess the threat posed by each CBV to the AV at each timestep. By employing this method, we eliminate the need for the viewpoint transformation

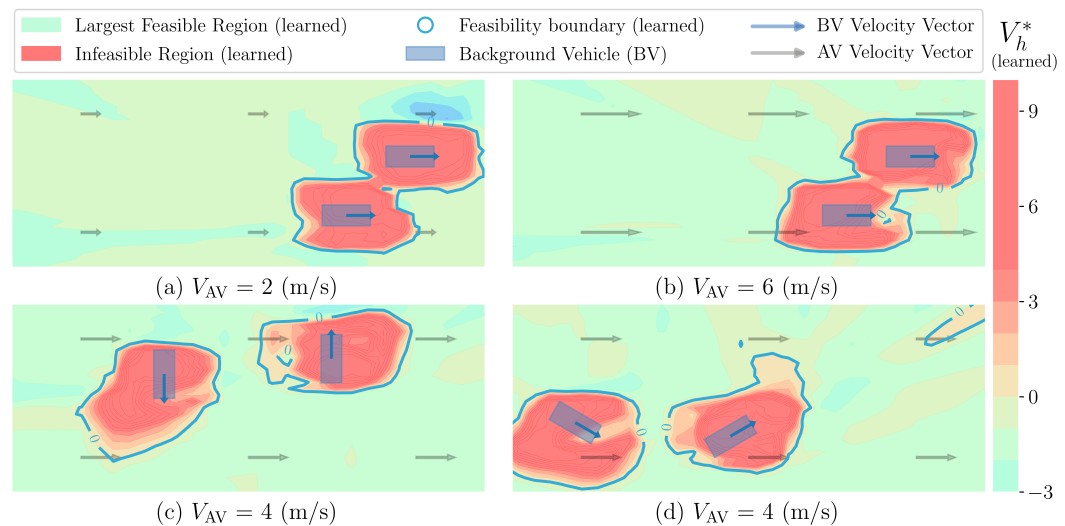

Figure 9: LFR visualization with 0.5m $d_{th}$ under various traffic scenarios.

Table 4: Detailed hyperparameters of feasible value function training.

| Parameter | Value |
|---|---|
| Optimizer | Adam ($\epsilon = 1e-5$) |
| Approximation function | Multi-layer Perceptron |
| Number of hidden layers | 2 |
| Number of hidden units per layer | 64 |
| Nonlinearity of hidden layer | RELU |
| Nonlinearity of output layer | linear |
| learning rate | Linear annealing $3e-4 \rightarrow 0$ |
| Batch size | 1024 |
| Training steps | 2e6 |
| Discount factor $\gamma$ | 0.98 |
| Expectile $\tau$ | 0.9 |
| Soft update | 5e-3 |
| $d_{th}$ | 0.1 |
| $M$ | 18 |

function $g(\cdot)$, previously described in Eqs. (6) and (9). Instead, the "pseudo" state is directly derived from the information available at each timestep, simplifying the process and enhancing the accuracy of LFR.

## D    Robustness of AV Pre-trained with Various CBV Methods

To explore the impact of reasonable adversarial scenarios on AV policy learning, we employed PPO [33] as the AV policy, which utilized the same forms of input and output as detailed for the CBV policy in Appendix B.2. By leveraging the AV's reward function described in [25], we trained the PPO policy across various surrogate CBV methods. Subsequently, we conducted robustness evaluations on these well-trained AV policies, presenting the results in Table 5.

As shown in Table 5, our proposed *FREA* method, when used as the CBV policy for AV training, improves the *Overall Score* (OS) and reduces the *Collision Rate* (CR), thereby enhancing AV policy robustness. This improvement primarily stems from the reasonable adversarial CBV policy. In normal traffic, the rarity of safety-critical scenarios limits AVs' ability to learn effective avoidance policies, especially for learning-based methods. Conversely, excessively adversarial policies like

Table 5: Comparative performance of AVs pretrained with various CBV methods across different maps. Results are the average of 10 runs in "Scenario9" with varied seeds.

| Surr. CBV | CBV | Town05 intersections | | | | | | Town02 intersections | | | | | |
|---|---|---|---|---|---|---|---|---|---|---|---|---|---|
| | | CR (↓) | OR (↓) | RF (↓) | UC (↓) | TS (↓) | OS (↑) | CR (↓) | OR (↓) | RF (↓) | UC (↓) | TS (↓) | OS (↑) |
| Standard | | 11% | 4m | 19m | 4% | 68s | 85 | **39%** | 8m | 20m | 18% | 57s | 70 |
| PPO | Standard | 17% | 11m | 17m | 6% | 66s | 82 | 40% | 11m | 19m | 15% | 63s | 70 |
| **FREA** | | **3%** | 6m | 18m | 1% | 75s | **89** | 40% | 3m | 18m | 19% | 56s | **71** |
| Standard | | 39% | 6m | 17m | 7% | 66s | 73 | 79% | 3m | 17m | 35% | 45s | 51 |
| PPO | **FREA** | 36% | 15m | 21m | 6% | 66s | 74 | 79% | 4m | 14m | 37% | 44s | 51 |
| **FREA** | | **31%** | 12m | 17m | 5% | 71s | **76** | **73%** | 3m | 20m | 28% | 51s | **55** |

PPO often result in unavoidable collisions, hindering the development of correct avoidance policies since no policy can ensure AV safety. By increasing the frequency of AV-feasible near-miss events, our *FREA* method provides effective data for AV training, thus improving policy robustness.

# E   Detail Explanation about the Metrics

## E.1   Near-miss Event Metrics

In traffic safety and vehicle dynamics, time-to-collision (TTC) and post-encroachment time (PET) are two commonly used metrics to quantify near-miss events[4]. Here are the formal definitions for these terms:

$TTC = \frac{d}{v_r}$, where $d$ represents the distance between the two vehicles or between a vehicle and an obstacle and $v_r$ is the relative velocity, which is the speed of one vehicle relative to the other or the obstacle.

$PET = t_2 - t_1$, where $t_1$ is when the first vehicle passes through the intersection point and $t_2$ is when the second vehicle passes through the same point.

In our work, the TTC metric was calculated using a publicly available code library [30], and the PET metric was computed using the code provided in [4].

## E.2   Feasibility Metrics

In Section 4.3, we introduce four key metrics to evaluate the feasibility of AV along collision trajectories induced by various CBV methods. Among these, the Infeasible Ratio (IR) and Infeasible Distance (ID) are novel metrics proposed in this paper. Their definitions and implications are detailed as follows:

**IR: Infeasible Ratio.** This metric quantifies the proportion of infeasible states for the AV along the collision trajectory induced by a specific CBV method. As the CBV approaches the AV, the likelihood of infeasibility naturally increases. The IR assesses the severity of collision risk by examining the percentage of these infeasible states, serving as a measure of the overall aggressiveness of the CBV collision trajectory. Since these trajectories inevitably lead to collisions, they must include segments where infeasible states occur. Therefore, evaluating the relative performance of IR is crucial.

**ID: Infeasible Distance.** This metric records the distance between the CBV and the AV when the AV first enters the infeasible region along the collision trajectory. Since CBVs may initiate their attacks from various starting points, the initial locations of collision trajectories can vary significantly. This variability means the IR metric alone may not fully capture the inevitability of a collision. The Infeasible Distance (ID) is therefore introduced to record the distance between the AV and CBV at the first instance when the AV's feasibility value exceeds zero. Intuitively, a larger ID indicates a more adversarial CBV approach, while a shorter ID suggests that the CBV allows the AV to remain feasible for longer, indicating a potentially avoidable collision.

Table 6: Detailed parameters of evaluation metrics.

| Metric | Weight $w^i$ | Maximum allowed value $m^i_{max}$ |
|--------|-------------|-----------------------------------|
| $CR$ | 0.4 | 1 |
| $OR$ | 0.1 | 10 (m) |
| $RF$ | 0.1 | 5 (m) |
| $UC$ | 0.3 | 1 |
| $TS$ | 0.1 | 30 (s) |

## E.3 Evaluation Metric of the AV

To fairly evaluate the performance of AV methods across different adversarial scenarios, we adopt the evaluation metrics from SafeBench [25], focusing on two main categories: *Safety level* and *Functionality level*. It is important to note that while the *Etiquette level* is discussed in the SafeBench paper, it has not been implemented in the actual code. This exclusion is likely due to the prioritization of safety and functionality in safety-critical scenarios, where etiquette is less essential. To avoid confusion, our evaluations strictly follow the implementation details specified in the SafeBench code, rather than the descriptions in the paper.

Within the *Safety level* and *Functionality level*, we define several specific metrics that contribute to an *overall score*, which is defined as a weighted sum of all evaluation metrics.

**Safety Level.** This level evaluates the safety performance of AV methods using two primary metrics: *collision rate* ($CR$) and *average distance driven out of road* ($OR$). We define $\tau$ as the scenario trajectory collected through interaction. The number of collisions in a scenario is represented by $c(\tau)$, and the distance driven out of the road is denoted as $d(\tau)$. Therefore, the metrics are calculated as follows: $CR = \mathbb{E}_\tau[c(\tau)]$, and $OR = \mathbb{E}_\tau[d(\tau)]$.

**Functionality Level.** This level measures the functional capabilities of AV agents in completing designated routes within testing scenarios. It employs three metrics: *route-following stability* ($RF$), *average percentage of uncompleted route* ($UC$), and *average time spent to complete the route* ($TS$). $RF$ quantifies the average distance between the AV and the reference route during testing, expressed as $RF = 1 - \mathbb{E}_\tau[\min\left\{\frac{x(\tau)}{x_{max}}, 1\right\}]$, where $x_{max}$ represents the maximum allowable deviation. $UC$ reflects the complement of the average completion percentage of the route, calculated as $UC = 1 - \mathbb{E}_\tau[p(\tau)]$, where $p(\tau)$ is the percentage of route completion of each testing scenario. $TS$ is defined as the average time required to complete a route, computed only for fully completed routes: $TS = \mathbb{E}_\tau[t(\tau)|p(\tau) = 100\%]$, where $t(\tau)$ denotes the time cost of each testing scenario.

**Overall Score.** The overall quality of AV methods is quantified by an *overall score* ($OS$), which aggregates the five metrics using a weighted sum formula: $OS = \sum_{i=1}^{5} w^i \times g(m^i)$, where each $m^i$ is a specific metric, $w^i$ is its weight, and $g(m^i)$ adjusts the metric based on its desirability:

$$g(m^i) = \begin{cases} \frac{m^i}{m^i_{max}}, & m^i \text{ is the higher the better} \\ 1 - \frac{m^i}{m^i_{max}}, & m^i \text{ is the lower the better} \end{cases}, \tag{20}$$

where $m^i_{max}$ is a constant representing the maximum allowed value for each metric $m^i$. Further details about $w^i$ and $m^i_{max}$ are provided in Table 6.

## F Scenario Analysis

### F.1 Successful Scenarios

To demonstrate that the *FREA* method can generate AV-feasible adversarial events across various traffic scenarios, we present additional visualizations from different towns and intersections, as shown in Figure 10. Specifically, Figure 10(a) illustrates scenarios where the AV is preparing to make a right turn, while the CBV exhibits adversarial behavior by overtaking and executing an early

right turn. Figure 10(b) displays situations where the CBV makes a U-turn from its lane, leading to a potential collision with the AV and creating adversarial scenarios. Figure 10(c) highlight the adversarial behavior of the CBV within an intersection, where CBV pre-empts the AV while passing through. These scenarios highlight the *FREA* method's adaptability to different traffic infrastructures and its effectiveness in generating safety-critical scenarios with reasonable adversariality.

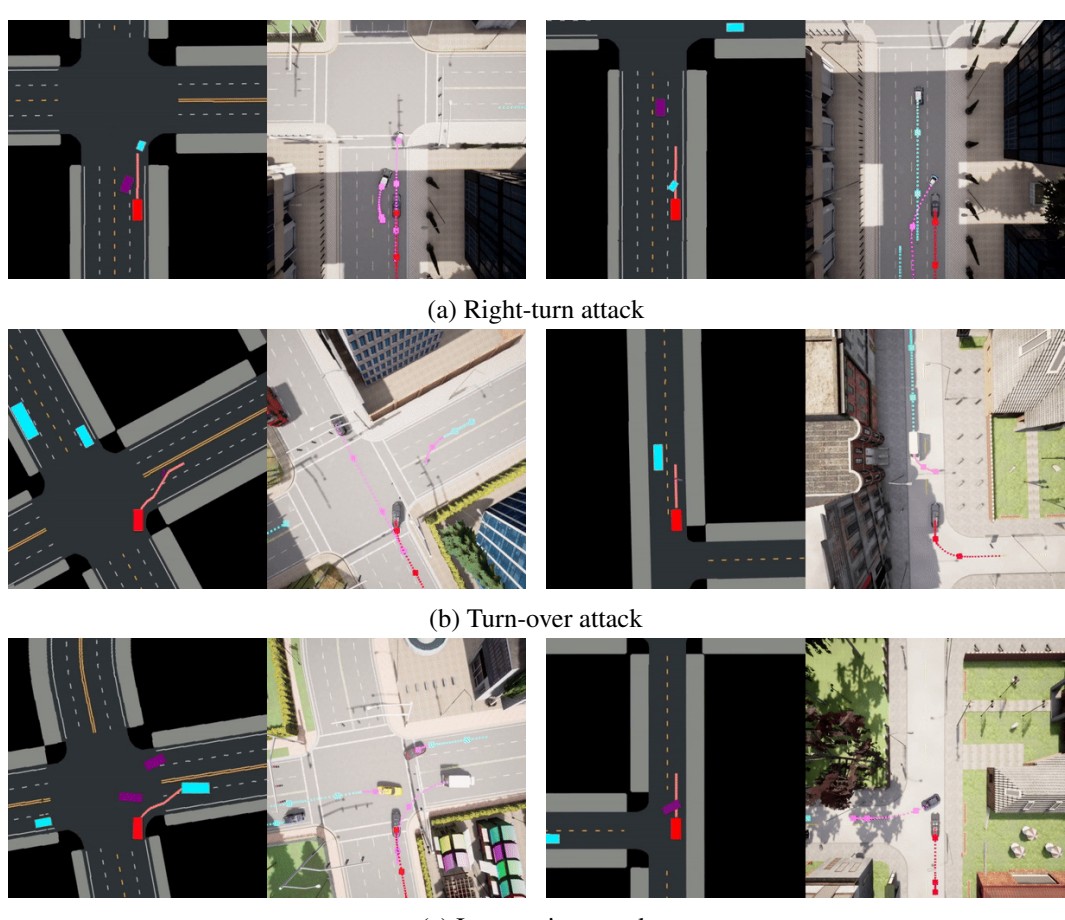

(a) Right-turn attack

(b) Turn-over attack

(c) Intersection attack

Figure 10: Successful scenarios. Red: AV (Expert). Blue: BV. Purple: CBV (*FREA*).

## F.2  Failure Scenarios

While the *FREA* method makes progress in generating adversarial yet reasonable safety-critical scenarios, particularly on the reasonableness of adversarial attacks, it also has several limitations, as illustrated by the failure cases depicted in Figure 11.

As illustrated in Figure 11(a), after completing its attack, the CBV is released as a normal vehicle on a narrow road. However, its turning radius exceeds the width of the road, forcing it onto the pavement and resulting in an unreasonable scenario. In Figure 11(b), after an unsuccessful initial attack due to the AV's obstacle avoidance maneuvers, the CBV attempts a second attack through a reverse maneuver. Although technically feasible, this approach deviates from typical driving objectives, compromising the scenario's reasonableness. Figure 11(c) shows a scenario where the CBV crosses a solid yellow line to initiate an attack. Since the state information in *FREA* lacks map details, violations of traffic rules are foreseeable. Future studies will focus on enhancing adherence to traffic regulations. Finally, Figure 11(d) depicts a traffic jam scenario at intersections caused by unreasonable attacks.

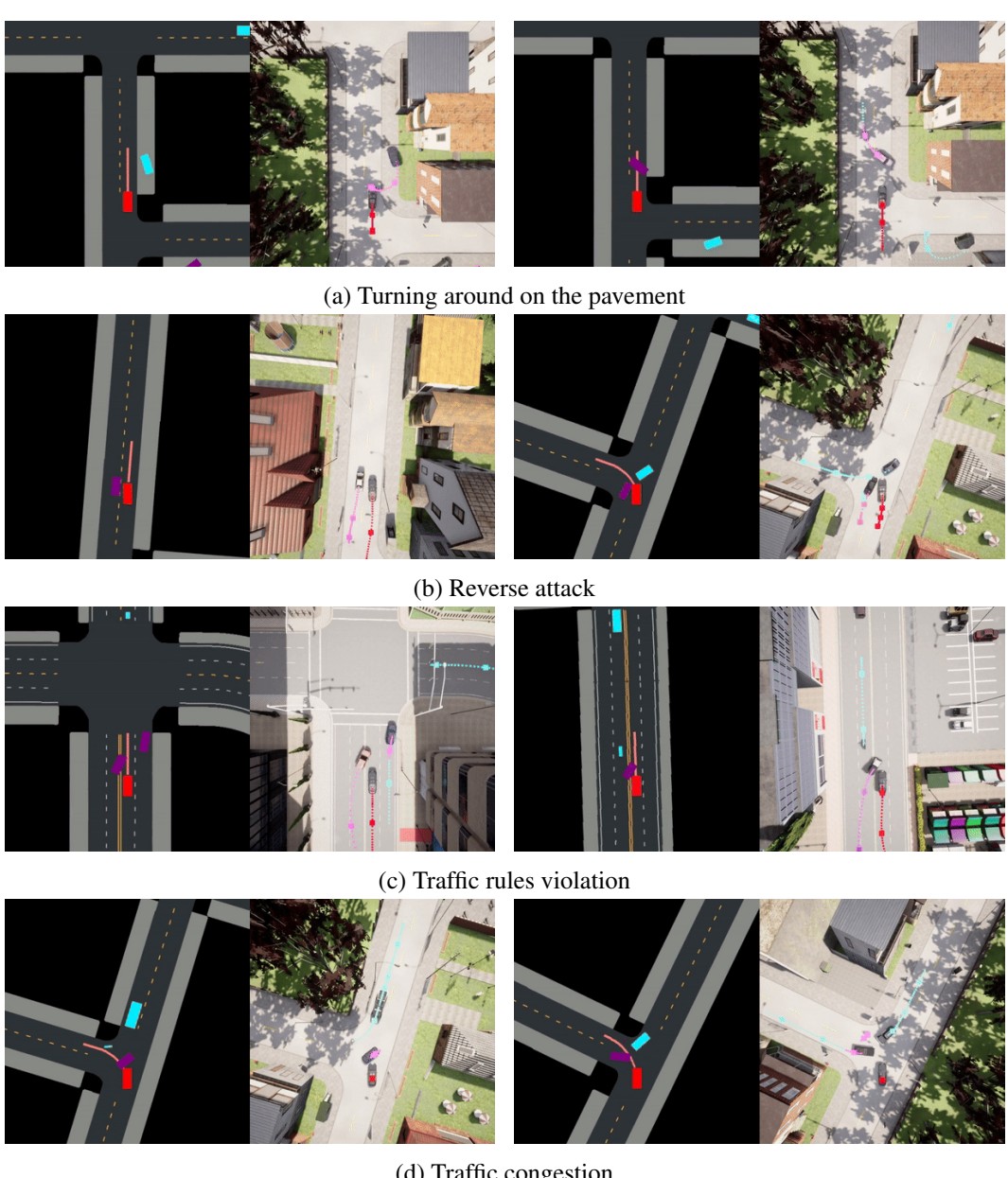

(a) Turning around on the pavement

(b) Reverse attack

(c) Traffic rules violation

(d) Traffic congestion

Figure 11: Failure scenarios: Red: AV (Expert). Blue: BV. Purple: CBV (*FREA*).

In conclusion, this paper acknowledges specific limitations in generating adversarial yet reasonable safety-critical scenarios. These limitations primarily include the lack of traffic regulation compliance and the challenge of aligning CBV behaviors during attacks with their driving objectives. Addressing these issues is crucial for developing more reasonable adversarial scenarios.

