# OpenReview forum: "FREA: Feasibility-Guided Generation of Safety-Critical Scenarios with Reasonable Adversariality"
_robot-learning.org/CoRL/2024/Conference — CoRL 2024_

### Official Review · Reviewer_oVnR · 2024-07-18
**Nice paper**

**Originality:** 4
**Technical Quality:** 4
**Clarity Of Presentation:** 3
**Potential Impact:** 3
**Recommendation:** 4
**Confidence:** 4

**Review:**

# Clarity

The clarity can be improved. A typically problem is the abuse of abbreviation. In each section the abbreviation need to be re-introduced. For example the LFR in Line 43 is not introduced in the introduction, but in abs.

# Originality

The paper is novel at combining the largest feasible region with constrained PPO.

# Strengths

1. The illustrative video is very impressive (but I have to say most of the collision is super unnatural).
2. The method is simple yet insightful. Combining Largest Feasible Region with constrained RL is promising.


# Weaknesses

1. The largest concern I have is the coverage of the offline dataset. The paper use Expert and Behavior agents (Line 459 in Appendix). I don't think those policies will generate sufficient violations and thus the estimation of the LFR might not be reliable.
2. As discussed in Appendix B.2, the reward for critical background vehicle is the reward encouraging it move toward the goal position of the autonomous vehicle. This naturally make the trajectories unrealistic. In the argument of previous work such as CAT (see Question 1), the reward encouraging collision might create unrealistic behavior. What's the purpose for creating unrealistic collisions?

**Quality Of The Limitations Section:**

3

**Questions For Rebuttal:**

1. Missing an important relevant work: CAT: Closed-loop Adversarial Training for Safe End-to-End Driving
2. The reward encouraging collision might create unrealistic behavior. (See weaknesses 2)

I don't have much question because the paper covers most of the details.

**Robotics Focus:**

1

**Summary Of Paper:**

Targeting the safety critical scenario generation, the paper first builds a value function measure the feasibility of the ego policy then uses that value function to form constraints and apply safe RL (constrained PPO) to find adversarial policy.

**Summary Of Recommendation:**

The proposed method is sound and promising in safety-critical scenario generation.

---

### Official Review · Reviewer_2ukq · 2024-07-22

**Originality:** 3
**Technical Quality:** 4
**Clarity Of Presentation:** 4
**Potential Impact:** 3
**Recommendation:** 3
**Confidence:** 4

**Review:**

**Strength**

The paper is well written and has covered most of the expected queries that arise while going through the different sections ranging from figures, limitations, metric attribution, ablations, etc.
The equations and the used notations are clearly defined.
It has good analysis and adaptation from the related papers.

The paper conducts a number of experiments that support its claim. It is able to demonstrate:
- with figure 2 that the trained LFR is reliable to be used for adversarial policy training,
- that FREA is capable of having more near-miss events than the general traffic flow and is also able to reduce over-collision events (infeasibility) due to the feasibility constraints as compared to other CBV methods.

The paper also introduces two new metrics called infeasible ratio and infeasible distance to evaluate the feasibility of AV along collision trajectories.

This possesses a nice application in learning robust and safe autonomous driving policies.

Although, this work does require some future research regarding the behavior of the learned policies of the AV and CBV to align them to some natural driving behaviors and traffic rules.


**Limitations**

- There were experiments to evaluate AV’s performance with adversarial scenarios. They show that safety (less collisions and less off-road) and progress improves. But with adversarial scenarios, it would be interesting to see the impact on any comfort related metrics as well. While the AV is trying to be safe and still make progress being robust enough in safety-critical situations, comfort would still be an important metric to optimize for provided the fact that the AV might be having some humans or some items that would prefer smoother trajectories. The goes for other violations as well.

**Quality Of The Limitations Section:**

3

**Questions For Rebuttal:**

**Questions and Suggestions**

Q1: Can we have a system/metric to check the chance/likelihood of the generated scenarios in the distribution of real-traffic scenarios? Ideally they should be in the long-tail of the distribution?

Q2: How can we keep the CBV policies or scenarios close to realistic driving or how are we preserving the naturalness of the generated scenarios?
- Was there an attempt to try to filter out the non-realistic CBV policies while training or an experiment to show how much such scenarios might affect the feasibility or robustness of the learned AV policies?

Q3: Line 103: “...under the proper assumption,...”: Could this be clarified? Is it the same as the assumption that the BVs follow deterministic policy or is it different?

Q4: Line 200: text should be “Figure 2(c) and (d)”

**Robotics Focus:**

3

**Summary Of Paper:**

The paper presents a method to generate reasonable safety-critical scenarios  with an upper-bound on the adversariality of the scenarios using the feasibility of the autonomous vehicle (AV). It calls the method FREA that uses the Largest **F**easibility Region (LFR) of the AV to  ensure the **RE**asonableness of the generated **A**dversarial scenarios. After calculating the LFR of the AV, FREA learns a feasibility-guided adversarial policy to control the critical-background vehicles (CBV) to generate adverse safety-critical scenarios.   The framework is inspired from the work of [FISOR (FeasIbility-guided Safe Offline RL)](https://arxiv.org/pdf/2401.10700). In a two stage framework:  FREA, first, uses offline data to learn an optimal feasible state-value network that is used to assess if the AV is within LFR.  With the LFR, FREA learns/optimizes (online) a feasibility-dependent adversarial objective to learn a policy for the CBVs that ensures AV’s feasibility.  The paper conducted experiments to demonstrate that FREA is capable of generating near-miss events (that represents reasonable adversariality) while maintaining the AV feasibility (that represents there exists policy that can ensure safety of the AV under maximum adversarial scenarios).

**Summary Of Recommendation:**

Adversarial scenarios are long-tail scenarios in real-traffic scenes and hard to scale.  This work would help in getting more robust self-driving policies trained on more and better adversarial scenarios. Overall, this framework to generate reasonable (maximum adversarial scenarios) that can still help to learn feasible policies can be helpful for future research using Safe-RL for autonomous driving. Therefore, I would recommend accepting this paper.

---

### Official Review · Reviewer_B6vA · 2024-07-22
**Presents a simple but effective idea for generating "near-miss" driving scenarios. Some opportunities for improvement.**

**Originality:** 3
**Technical Quality:** 3
**Clarity Of Presentation:** 3
**Potential Impact:** 3
**Recommendation:** 3
**Confidence:** 4

**Review:**

This paper is reasonably well-written, technically sound, offers an original solution to a relevant problem, and will be of use to the robotics / AV community.

Strengths:
- Scalable solution to real-world problem
- strong results against baseline

Weaknesses:
- Exposition / clarity
- lacking analysis of method in more general scenarios, or evaluating quality of LFR approximation.

The main weakness with regards to clarity is in the development of some of the methodology, specifically with regards to section 3.2. Specific comments regarding this are listed below.
As for the analyses, I would like to see a more thorough analysis of the LFR method. Specifically, the learned LFRs are evidently noisy. It would be good to compare these to those that are computed from HJ reachability, for example (for e.g. the small scale examples explored in Fig. 2). It would also be good to understand how the inaccuracy in the LFR approximation manifests in the resulting behavior of the BVs.


Comments/questions:
- Contribution 2 is same as 1?
- (locally) Jointly optimizing for N agents doesn’t suffer from curse of dimensionality. Can be performed at roughly same cost as N* cost of optimizing for 1 agent.  e.g. see GNEP etc. It appears your reference doesn’t discuss this point anyways but is rather talking about rare events in high dimensional data/strategy space.
- Constrained/Safe RL: probability of constraint violation is zero? defining function on random variable => output of h(x) is random variable, not meaningful to constrain that
- Same in definition 1. Assuming your policy is stochastic, (as is indicated by ~ symbol) then unless h maps from distributions over state space to reals, this is not meaningful definition
- 130 what is “optimal” feasible value function? The value function simply is what it is, there is no notion of optimality
- 131 minimizing over what? Parameterization of the functions V/Q? not specified
- This optimization procedure/ dataset assumes that S_0^{AV} is known and fixed for all datapoints?
- S^{AV} is overloaded. Training example points but also “current state” of AV at some query point.
- Section 3.2 is not very clear on how CBV policies are actually generated. What is the “progress” objective?  Is goal to traverse to zero level set?
- Lemma 1, not clear why a^{AV} = g(s’) for some s’? Need Lemma 1 to stand alone without referencing to appendix.
- Equation 5 and 6 motivation is not presented clearly
- Figure 2a, why is LFR region at rear of obstacles not flat/symmetric?  How accurate are these LFRs actually, compared to e.g. those computed by HJ reachability?
- Define TTC and PET formally

**Quality Of The Limitations Section:**

2

**Questions For Rebuttal:**

- Contribution 2 is same as 1?
- (locally) Jointly optimizing for N agents doesn’t suffer from curse of dimensionality. Can be performed at roughly same cost as N* cost of optimizing for 1 agent.  e.g. see GNEP etc. It appears your reference doesn’t discuss this point anyways but is rather talking about rare events in high dimensional data/strategy space.
- Constrained/Safe RL: probability of constraint violation is zero? defining function on random variable => output of h(x) is random variable, not meaningful to constrain that
- Same in definition 1. Assuming your policy is stochastic, (as is indicated by ~ symbol) then unless h maps from distributions over state space to reals, this is not meaningful definition
- 130 what is “optimal” feasible value function? The value function simply is what it is, there is no notion of optimality
- 131 minimizing over what? Parameterization of the functions V/Q? not specified
- This optimization procedure/ dataset assumes that S_0^{AV} is known and fixed for all datapoints?
- S^{AV} is overloaded. Training example points but also “current state” of AV at some query point.
- Section 3.2 is not very clear on how CBV policies are actually generated. What is the “progress” objective?  Is goal to traverse to zero level set?
- Lemma 1, not clear why a^{AV} = g(s’) for some s’? Need Lemma 1 to stand alone without referencing to appendix.
- Equation 5 and 6 motivation is not presented clearly
- Figure 2a, why is LFR region at rear of obstacles not flat/symmetric?  How accurate are these LFRs actually, compared to e.g. those computed by HJ reachability?
- Define TTC and PET formally

**Robotics Focus:**

3

**Summary Of Paper:**

This paper presents a method for generating near-miss driving scenarios. The method involves precomputing a representation of the LFR, then leveraging constrained RL to train environmental agents to approach Ego vehicle but not force collision.

**Summary Of Recommendation:**

Weak accept. Good method, but some additional analyses would help, and improving clarity of the methodology is needed.

---

### Author Rebuttal · Authors · 2024-08-07

This is our revised manuscript.

---

### Decision · Program_Chairs · 2024-09-04

**Decision:**

Accept

**Comment:**

The paper presents a method to generate near-miss driving scenarios while maintaining the AV feasibility.

Strengths
- The approach is scalable to the real-world complexity of autonomous driving.
- The method performs well compared to existing baselines by capturing more near-miss events than the general traffic flow and by reducing the infeasible scenarios as compared to other CBV methods.

Limitations
- The paper needs to comment on the reliability of the LFR. The learned LFR can be noisy. It is important to do a proper analysis of how that affects the ability to generate near-miss scenarios.
- The CBV policies can result in unrealistic driving behaviors.  How does this affect the naturalness of the generated scenarios?
- The paper needs to improve its exposition and clarity.

**Comments post rebuttal**

The revised version has improved the paper's clarity. The reviewers also appreciate the inclusion of comfort-related metrics and the effect of extreme adversarial CBVs on the AV policies, as well as additional results on the stability of LFR. The authors should incorporate the results from the rebuttal in the final paper.